# Effects of Salinity Stress on Histological Changes, Glucose Metabolism Index and Transcriptomic Profile in Freshwater Shrimp, *Macrobrachium nipponense*

**DOI:** 10.3390/ani13182884

**Published:** 2023-09-11

**Authors:** Yiming Li, Yucong Ye, Wen Li, Xingguo Liu, Yunlong Zhao, Qichen Jiang, Xuan Che

**Affiliations:** 1Fishery Machinery and Instrument Research Institute, Chinese Academy of Fisheries Sciences, Shanghai 200092, China; liyiming183@163.com (Y.L.); liuxingguo@fmiri.ac.cn (X.L.); 2School of Life Science, East China Normal University, Shanghai 200241, China; ycy_666@yeah.net (Y.Y.); 10191910010@stu.ecnu.edu.cn (W.L.); ylzhao426@163.com (Y.Z.); 3Freshwater Fisheries Research Institute of Jiangsu Province, Nanjing 210017, China; qichenjiang@live.cn

**Keywords:** salinity stress, transcriptome analysis, histological change, glucose metabolism, *Macrobrachium nipponensis*

## Abstract

**Simple Summary:**

Salinity has great influences on ion homeostasis and the physiological activities of crustaceans in aquatic environment. Transcriptome analysis of *Macrobrachium nipponense* showed that differentially expressed genes were mainly related to ion homeostasis, glucose metabolism and lipid metabolism. In addition, the morphological structure of *M. nipponense* gill tissue under high salinity stress showed significant changes in gill filaments, gill cavities and mucosal structures. Our study showed that salinity stress activates the ion transport channel of *M. nipponense* and promotes the up-regulation of glucose metabolism, and that high salinity causes damage to the gill tissue structure of *M. nipponense*.

**Abstract:**

Salinity is an important factor in the aquatic environment and affects the ion homeostasis and physiological activities of crustaceans. *Macrobrachium nipponense* is a shrimp that mainly lives in fresh and low-salt waters and plays a huge economic role in China’s shrimp market. Currently, there are only a few studies on the effects of salinity on *M. nipponense*. Therefore, it is of particular importance to study the molecular responses of *M. nipponense* to salinity fluctuations. In this study, *M. nipponense* was set at salinities of 0, 8, 14 and 22‰ for 6 weeks. The gills from the control (0‰) and isotonic groups (14‰) were used for RNA extraction and transcriptome analysis. In total, 593 differentially expressed genes (DEGs) were identified, of which 282 were up-regulated and 311 were down-regulated. The most abundant gill transcripts responding to different salinity levels based on GO classification were organelle membrane (cellular component), creatine transmembrane transporter activity (molecular function) and creatine transmembrane transport (biological function). KEGG analysis showed that the most enriched and significantly affected pathways included AMPK signaling, lysosome and cytochrome P450. In addition, 15 DEGs were selected for qRT-PCR verification, which were mainly related to ion homeostasis, glucose metabolism and lipid metabolism. The results showed that the expression patterns of these genes were similar to the high-throughput data. Compared with the control group, high salinity caused obvious injury to gill tissue, mainly manifested as contraction and relaxation of gill filament, cavity vacuolation and severe epithelial disintegration. Glucose-metabolism-related enzyme activities (e.g., pyruvate kinase, hexokinase, 6-phosphate fructose kinase) and related-gene expression (e.g., *hexokinase*, *pyruvate kinase*, *6-phosphate fructose kinase*) in the gills were significantly higher at a salinity of 14‰. This study showed that salinity stress activated ion transport channels and promoted an up-regulated level of glucose metabolism. High salinity levels caused damage to the gill tissue of *M. nipponense*. Overall, these results improved our understanding of the salt tolerance mechanism of *M. nipponense*.

## 1. Introduction

Salinity has an obvious effect on the osmotic pressure of aquatic animals. It can fluctuate with weather, precipitation, human disturbance and other factors [1,2,3]. Osmotic pressure can change in organisms along with changes in environmental osmotic pressure due to factors such as temperature, salinity and pH [4,5,6]. The mechanism of osmotic pressure regulation in aquatic animals has always been an intense topic of global research. In the process of evolution, most aquatic animals have formed a set of systems to maintain osmotic pressure balance, including organs of osmotic pressure regulation [7], changes of serum ions [8], ion transport enzymes [9], etc. This system can ensure that aquatic animals can adjust their osmotic pressure within a certain salinity range to maintain normal vital signs [10].

With the development of the global aquaculture industry, achieving higher economic benefits and use of resources, desalination aquaculture of seawater varieties and salinity domestication of freshwater varieties have become important research directions in some countries and regions [11]. The optimization of the farming mode is beneficial not only to the development of the industry, but also to the rational utilization of water resources [12]. Studies have shown that there are 36 million square hectares of inland saline–alkali land distributed in various regions of China, and the sustainable use of such land has always been the focus of researchers from diverse fields [13]. For the aquaculture industry, salinity domestication of some salt-tolerant varieties is not only conducive to the use of water resources in saline–alkali land, but also conducive to the development of aquaculture. Li et al. [14] studied the salt tolerance of juvenile Malaysian Red Tilapia (*Oreochromis mossambicus* × *O. niloticus*) and found that acute salinity stress was not conducive to their survival. However, after salinity domestication, Malaysian Red Tilapia were able to survive in a highly saline environment. They have a very strong salt tolerance and are a potential species for saline–alkali land culture [15]. The survival rate of economic species such as Red Tilapia [16] and *Paralichthys olivaceus* [17] decreases significantly after acute salt stress. It can be seen that a change in salinity is a very important factor for the survival of aquatic organisms.

With the development of omics sequencing technology, transcriptomic, proteomic and metabolomic technologies have become new technical methods to study the rules of various life activities in aquatic animals [18]. Among these, transcriptome technology is used to study the real-time transcripts of certain tissues or organs under specific stimuli [19]. Transcriptomic techniques have also been used widely in the study of crustaceans coping with salinity stress, and have been applied in *Exopalaemon carinicauda* [20], *Eriocheir sinensis* [21] and *Penaeus monodon* [22]. Studies have shown that the expression of anthocyanin and rhodopsin genes can help shrimp to cope with salinity stress [23]. In addition, studies have shown that the renin-angiotensin system (RAS) plays important roles in salinity adaptation in *Penaeus vannamei* [24]. At present, there are more and more transcriptomic studies being conducted on the response of aquatic organisms to saline environments. We chose *M. nipponense* as our research object in order to explore the transcriptomic response of aquatic organisms to a saline water environment.

Aquatic animals respond to changes in environmental salinity by regulating of osmotic pressure. Both aquatic vertebrates and invertebrates have organs, tissues or cells that function in osmotic pressure regulation [6]. Gill tissue is in direct contact with the external environment of aquatic animals and is the main organ for filter feeding and gas exchange [25]. Gill tissue is also important in ion exchange, which plays a crucial role in maintaining osmotic pressure balance [26]. When gills encounter external environmental stimulation, they will cause changes in energy metabolism to maintain homeostasis [27].

As an important part of energy metabolism, glucose metabolism plays an important role in coping with external pressure and maintaining homeostasis [28]. Pyruvate kinase (PK) is involved in glycolysis, catalyzing the formation of a second ATP reaction [29]. Hexokinase (HK) is one of the key enzymes in the glycolytic pathway and plays an important role in the regulation of glucose metabolism [30]. Allosteric enzyme 6-phosphofructokinase (PFK) is involved in the glycolytic pathway and acts as a rate-limiting enzyme [31]. Lactate dehydrogenase (LDH) is involved in the regulation of anaerobic glycolysis and gluconeogenesis [32]. It can participate in the catalytic process of reductive oxidation between propionic acid and L-lactic acid, as well as related alpha-ketoacids. These four enzymes are closely related to the glucose metabolism of organisms. Therefore, in this study, the enzymic activity of PK, HK, phosphate fructose kinase (PFK) and LDH, as well as their gene expression levels, were detected to reflect the osmotic regulation of *M. nipponense* under different salinities.

*M. nipponense* is a freshwater shrimp (phylum Arthropoda, subphylum Crustacea and order Decapoda) that is mainly distributed in tropical and subtropical areas of the world [33,34,35]. It has a delicious taste and is favored by consumers, so it has a high economic value [36]. The annual output of artificial breeding has exceeded 200,000 tons, with an output value of USD 27.5 million [37]. It has been reported that *M. nipponense* probably originated from the ocean, moved from seawater to fresh water and successfully adapted to a freshwater environment [34]. In recent years, most studies have focused on nutritional regulation [38], germplasm resource investigation [37] and immune performance [39], but there are fewer studies on the mechanism of metabolic physiology at different salinities. Previous studies have reported that, under acute stress, adult *M. nipponense* could live normally in a salinity range of 7–20 [40]. Transcriptome sequencing in this study showed an increased expression of genes associated with ion transport and glucose metabolism. A salinity of 14‰ promoted glucose metabolism in *M. nipponense*, and high salinity levels caused damage to gills. Our findings provide a theoretical basis for the physiological response mechanism to salinity in *M. nipponense*.

## 2. Material and Methods

### 2.1. Experimental Organism

Juvenile *M. nipponense* shrimp were purchased from Shanghai Xiangsheng Fisheries Cooperative (Caojing, Jinshan, Shanghai). After 1 week of temporary rearing under laboratory conditions, the *M. nipponense* were subjected to salinity adjustment, in which one group served as a freshwater control group. The salinity levels of the other groups were set at 8 ± 0.1‰, 14 ± 0.1‰ (isotonic group) and 22 ± 0.1‰, and these groups were adapted to the salinity conditions using a gradual salt addition method. When salinity adjustment was complete, shrimp (initial body weight was 1.84 ± 0.12 g) were acclimated to the salinity and stabilized for 1 week, followed by a 6-week salinity culture experiment in 90 L white plastic tanks. During the experiment, the water temperature was 25 ± 1 °C, the pH was 7.6 ± 0.5, and the dissolved oxygen in the water was ensured to be above 6.2 ± 0.6 mg/L. Commercial feed was fed twice a day at 8:00 and 18:00, and the water was changed at 1/4 of the day. The brine of the desired salinity was prepared in advance and continuously aerated for more than 24 h. Feeding was suspended 24 h before the end of the experiment, and nine *M. nipponense* were randomly selected from each group. The gill tissues were sampled for subsequent histology, enzyme activity, gene expression analysis and transcriptome sequencing. Three of these gill tissues were used for histological analysis, three for enzyme activity and gene expression analysis and the last three for transcriptome sequencing. Gill tissues from each salinity group were also taken for liquid nitrogen flash freezing and then stored at −80 °C.

### 2.2. RNA Extraction and Illumina HiSeq Sequencing

The gill tissues (three biological replicates) from the control (C) and isotonic groups (14 ± 0.1‰) (T) were selected for transcriptome sequencing. In our previous studies, we found that the S14 group was the isotonic point and the growth performance in all aspects was better at this salinity level [41]. The total RNA from the gill tissues of *M. nipponense* was extracted using the TRIzol reagent (Invitrogen, Waltham, MA, USA) according to the manufacturer’s protocol. Total RNA extraction was followed by gel electrophoresis for quality control and concentration determination using a NanoDrop 2000 (Thermo Fisher Scientific, Waltham, MA, USA). Total RNA of acceptable quality (2.0 ≥ 28 S/18 S ≥ 1.8) was selected for library construction. Subsequently, magnetic beads containing Oligo (dT) were used to enrich the mRNA in the total RNA, and then the cleaving agent was added to cleave the mRNA into small fragments. A cDNA chain was synthesized by 6-base random primers, and then double-strand cDNA was synthesized by a double-strand synthesis reaction system and purified using a kit. The end of the purified double-stranded cDNA was repaired, the A-tail connector was added, the fragment size was selected and PCR was used to amplify the fragment. The library was examined using the Agilent 2100 bioanalyzer (Agilent Technologies Inc., Santa Clara, CA, USA) and sequenced using the Illumina HiSeqTM 2500 sequencer from Shanghai Ouo Biotechnology Co., Ltd. (Shanghai, China).

### 2.3. Transcriptomic Assembly and Functional Annotation

Raw reads containing poly-N and low-quality reads were trimmed in Trimmomatic version 0.39 to obtain clean data [42]. The transcript sequences were obtained by paired-end assembly of Trinity software (version: trinityrnaseq_r20131110). Considering the sequence similarity and length conditions, the longest splicing sequence was selected as a single gene. TGICL (version: 2.1) software was used to cluster and remove redundancies, and the final obtained single genes were extended to obtain a set of single genes for subsequent analysis. The obtained unigene was compared in the non-redundant protein database (NR), Clusters of Orthologous Groups of proteins (COG), Gene Ontology (GO), Swissprot, evolutionary genealogy of genes: Non-supervised Orthologous Groups (eggNOG) and Kyoto Encyclopedia of Genes and Genomes (KEGG) databases using diamond software, and HMMER 3.3.2 software and Pfam databases were used to perform unigene functional analysis. The FPKM (fragments per kb of transcript per million reads) method was used to calculate the number of specific gene fragments per kilobase length per million reads to represent single gene expression. The FPKM calculation formula was as follows:$$\mathrm{FPKM}\left(A\right)=\frac{\mathrm{Number}\;\mathrm{of}\;\mathrm{fragments}\;\mathrm{compared}\;\mathrm{to}\;\mathrm{gene}\;A}{\mathrm{Comparison}\;\mathrm{to}\;\mathrm{the}\;\mathrm{number}\;\mathrm{of}\;\mathrm{fragments}\;\mathrm{of}\;\mathrm{all}\;\mathrm{genes}\;\times \;\mathrm{length}\;\mathrm{of}\;\mathrm{gene}\;A}{\text{×}10}^{9}$$

### 2.4. Differentially Expressed Gene (DEG) Comparison and Function Enrichment

The number of counts for each sample unigene was normalized (based on the mean to estimate expression) using DESeq (3.17) software. The differential multiplicity was calculated and the number of reads was tested for the significance of differences using the negative binomial distribution test (NB). Finally, considering the test results of multiplicity and significance of difference, the differential expression of this single gene was screened. The fold change and *p*-value or false discovery rate (FDR, adjusted *p*-value) were used to analyze whether the same unigene was differentially expressed in two samples, and the default conditions for screening differences were a *p*-value < 0.05 and a difference ploidy > 2.

### 2.5. Histological Analysis

Histopathological analysis was performed with reference to Wang et al. (2022) [43]. First, gill tissues that were collected from *M. nipponense* reared for 6 weeks in each salinity group were immersed in 4% paraformaldehyde solution for over 48 h for fixation. Then, they were dehydrated with multiple groups of ethanol solutions of different concentrations for 30 min each (50%, 70%, 80%, 90%, 95% and 100%). The tissue was soaked in xylene for 40 s and then transferred to paraffin for 3 h to complete the embedding. Finally, tissue sections of 5 µm thickness (cross-section) were performed using a German Leica RM2235 sectioning machine and stained with hematoxylin and eosin (H&E). The stained tissue sections were sealed with a neutral adhesive and then observed and photographed with an Olympus BX51 optical microscope (Olympus Corporation, Tokyo, Japan). Three samples from each group were taken for the histological examination of the gills.

### 2.6. Validation of Transcriptomic Data and Glucose Metabolism-Related Gene Expression by qRT-PCR

To verify the transcriptome assembly results, 15 DEGs were selected for qRT-PCR analysis. TRIzol reagent (Accurate Bio, Changsha, China) extraction of total RNA in gill tissues was used to for transcriptome validation and to analyze the gene expression of glucose metabolism (HK, PK, G-6-Pase and LDH). Primer Premier 5 software was used for primer design, and the primer sequences are shown in Table 1 and Table 2. The reference gene was β-actin. The reaction mixture (20 μL) contained 2 × ChamQ Universal SYBR qPCR Master Mix 10 μL, 1 cDNA template and 0.4 μL (10 μM) of each of the forward and reverse primers. The setting expansion program was as follows: initial degeneration for 30 s, 95 °C; a 95 °C cycle for 5 s; and a 60 °C cycle for 20 s. The specificity of the amplified product was confirmed by melting curve analysis. The relative mRNA expression of each gene was calculated using the comparative CT method 2^−ΔΔCt^ (Fan et al., 2022 [44]).

### 2.7. Determination of Glycometabolic Enzyme Activities

Samples stored at −80 °C were removed and placed on ice. Pre-cooled 0.9% sterile saline was added to gill tissue (0.1 g) from each *M. nipponense* group at a ratio of 1:9 and homogenized in an ice water bath for 90 s. They were quickly placed at 4 °C and centrifuged at 3500× *g* rpm for 15 min. The tissue supernatant was then transferred to a new centrifuge tube to prepare the tissue stock solution. The enzyme activity assay kits used in this experiment were purchased from the Nanjing Jiancheng Institute of Biological Engineering (Nanjing, China). The enzyme activities related to glucose metabolism were measured by a PK assay kit (A076-1-1), an HK assay kit (A077-3-1), a phosphofructokinase (PFK) assay kit (A129-1-1) and a lactate dehydrogenase (LDH) kit (A020-2-2). Gill tissues were assayed for protein content before the enzyme activity assay using the Thomas Brilliant Blue method. All operations were performed according to the manufacturer’s instructions.

### 2.8. Statistical Analysis

The Shapiro–Wilk test was used to check the normality of all data, and Levene’s test was used to check the homogeneity of variance. A one-way analysis of variance (ANOVA) was used for data analysis. If the difference was significant, the Tukey multiple comparison test was used. SPSS 20.0 software (IBM, New York, NY, USA) was used for statistical analysis. When *p* < 0.05, the difference was statistically significant.

## 3. Results

### 3.1. Basic Growth Index

After 6 weeks of the salinity culture experiment, the growth indices of *M. nipponense* in different experimental groups were measured, and the results are shown in Table 3. The survival rate of *M. nipponense* in salinity group 14 was the highest, while that in salinity group 22 was the lowest. The weight gain rate and hepatosomatic ratio of *M. nipponense* in salinity group 14 were the highest, and were significantly different from those in the control group (*p* < 0.05), while brachium nipponense with salinity 22 had the lowest weight gain rate, and there was no difference in hepatosomatic ratios between the other groups (*p* > 0.05).

### 3.2. Transcriptome Sequencing and Assembly

The transcriptome sequencing results are shown in Table 4, and the raw data were uploaded to NCBI (accession number: PRJNA941036). The total numbers of raw paired-end reads for the control group were 49,299,632, 49,394,440 and 49,316,632, and for the isotonic salinity group (S14) group these were 48,933,058, 48,934,960 and 48,973,892. After removing low-quality reads, the results showed that the effective base rates of the six libraries were 95.30% (7,047,263,043 bp), 95.19% (7,052,820,332 bp), 95.33% (7,052,349,647 bp), 95.50% (7,009,780,552 bp) and 95.46% (7,007,194,602 bp) and 95.55% (7,018,985,183 bp), respectively. The Q30 percentage and GC content for the entire data set were higher than 95% and 46%, respectively.

### 3.3. Analysis of Differentially Expressed Genes (DEGs)

The DEGs are shown in Figure 1. After principal component analysis (PCA) of the gene expression dataset, it was found that the molecular pathways and corresponding gene expressions were significantly different between the control and S14 groups (Figure 1A). A total of 593 DEGs were identified (FDR, ≤0.01; fold-change, ≥2), of which 282 were up-regulated and 311 were down-regulated (Figure 1B–D). Looking at the heat map, it can be found that the S14 group expressed different genes than the control group (Figure 2).

### 3.4. GO and KEGG Enrichment Analysis

We performed KEGG pathway and GO functional enrichment to further explore the potential role of DEGs between the control group and the S14 group. Overall, 593 DEGs were mapped to three categories of GO, which included biological process (396), cellular component (277) and molecular function (292). In terms of biological processes, the DEGs mainly included “creatine transmembrane transport” (GO: 1902598), “amino acid transport” (GO: 0006865) and “defense response to bacterium” (GO: 0042742). For cellular components, DEGs mainly affected “organelle membrane” (GO: 0031090), “integral component of plasma membrane” (GO: 0005887) and “extracellular region” (GO: 0005576). In molecular functions, the DEGs mainly affected “creatine transmembrane transporter activity” (GO: 0005308), “monocarboxylic acid transmembrane transporter activity” (GO: 0008028) and “RNA-directed DNA polymerase activity” (GO: 0003964) (Figure 3).

As shown in Figure 4, through enrichment analysis of the KEGG pathway, we identified the 20 most significant pathways related to salinity (*p* < 0.05). (Figure 4). Under different salinities, the most significant pathways were the “AMPK signaling pathway” (ko04152), “Lysosome” (ko04142) and “Drug metabolism-cytochrome 450” (ko00982).

### 3.5. Validation of DEG and Glucose Metabolism-Related Gene Analysis

We performed real-time fluorescent quantitative PCR analysis for 15 genes (Figure 5) to validate the transcriptome data. The results showed that the expression pattern was consistent with that of RNA-seq, and the correlation coefficient reached 0.96 (R = 0.96), indicating that RNA-seq was accurate and reliable.

As shown in Figure 6, we analyzed related genes to explore the effect of salinity on glucose metabolism in the gills of *M. nipponense*. With an increase in salinity, the expression of *HK*, *PK* and *G-6-Pase* showed an overall trend of first increasing and then decreasing. The expression reached its highest point in S14; expression in the 22‰ salinity (S22) group was inhibited and was significantly lower than that in the control group (*p* < 0.05) (Figure 6A–C). There was no significant difference in *LDH* expression in the control group, and the 8‰ salinity (S8) and S22 groups had significantly higher *LDH* expression than the control group (*p* < 0.05) (Figure 6D).

### 3.6. Histological Analysis of Gills

As shown in Figure 7, the effect of salinity on the gill structure of *M. nipponense* was studied. In Figure 7A, it can be seen that the epidermis of the outer horn of the gill tip of *M. nipponensis* in the control group was well-shaped, the epithelium of the gill plate appeared was slightly distorted and the basement membrane and the overall structure were intact. As shown in Figure 7B, compared with the control group, the epidermis of the outer horn of the gill tip of *M. nipponensis* in the S8 group was swollen and distorted, the epithelium of the gill plate was distorted with the basement membrane and the overall structure was similar to that of the control. In Figure 7C, the gill lamellae in the S14 group were evenly spaced and arranged neatly compared with the control group, there was no secretion between the gill lamellae, columnar cells were evenly arranged, connective tissue connections were normal and the structure of the gill tissue was intact. In Figure 7D, compared with the previous groups, the gill plate gap in group S22 began to increase significantly, and the gill epithelium and basement membrane were completely broken and disintegrated. Disintegrating columnar cells were also observed in the space between the branchial plates. Most of the connective tissue was broken and disappeared, vacuoles appeared, and the gill tissue structure was greatly damaged.

### 3.7. Glucose Metabolism-Related Enzyme Activity Analysis

As the salinity increased, the PK, HK and PFK activity showed a trend of first increasing and then decreasing, and was highest in the S14 group (Figure 8A–C). In the S22 group, PK and PFK activities were significantly lower than those in the control group (*p* < 0.05), while there was no significant difference in HK activities (*p* > 0.05). For LDH activity, the S14 group showed significantly lower expression than the control group (*p* < 0.05), whereas the S8 and S22 groups showed significantly higher expression than the control group (*p* < 0.05) (Figure 8D).

## 4. Discussion

Salinity has a great influence on the osmotic pressure of aquatic animals. In addition, changes in salinity can also affect the growth, development and other physiological reactions of aquatic animals [4,25]. When aquatic animals are in a salinity-stressed environment, they will consume energy for osmotic regulation, which can account for 20–50% of the total energy consumption [45]. *M. nipponense* is an aquaculture species with high economic value. For aquaculture, the domestication of some salt-tolerant varieties is not only conducive to the utilization of saline–alkali land water resources, but also to the development of the aquaculture industry [13,46]. In recent years, transcriptome technology has gradually matured, and the cost of sequencing has been continuously reduced. Transcriptomics has been applied more and more widely in the process of revealing the changes to molecular signals in aquatic animals during changes in life activities [16,19]. In our study, a total of 593 different genes were screened by transcriptome sequencing, including 282 up-regulated genes and 311 down-regulated genes. They are mainly involved in the AMPK signaling pathway (ko04152), lysosome (ko04142) and drug metabolism-cytochrome 450 (ko00982). We conducted further fluorescence quantitative verification of 15 differential genes.

We found that most up-regulated genes in the fluorescence quantitative verification results were related to ion transport and glucose metabolism. Studies have shown that cytochrome p450 is involved in energy metabolism and oxidative stress of shrimp under salt stress, and plays an important role in the salinity adaptation of *L. vannamei* [47]. The Na^+^/K^+^ symporter can be used as an index of the active transport of sodium and potassium ions and energy dissipation by osmotic pressure regulation [48]. The change in its enzyme activity may be closely related to the change in energy metabolism during salinity acclimation [48]. In the process of transporting amino acids, sodium and chloride-dependent glycine transporter and sodium-dependent proline transporter-like transport ions were found both in the same direction and in the opposite direction [49,50]. Up-regulation of these genes increases the rate of ion transport, a process that requires energy. Sulfatase is involved in the metabolic process of the sulfated sugar chain through desulfurization and acidification [51]. After the cell produces lactic acid via glycolysis, it is excreted from the cell via a monocarboxylate transporter [52]. Fucosyltransferase catalyzes important intermediates of carbohydrate metabolism to participate in fucosylation [53]. Mannosyl converts mannose in the glycolysis pathway to participate in energy metabolism [54]. C-type lectins are a class of calcium-dependent sugar-binding proteins [55]. The up-regulation of these genes indicates that under isotonic conditions, the proportion of energy provided by glucose metabolism for osmotic regulation of *M. nipponense* increased. The comparison of differential genes in the KEGG database showed 20 of the most significant salinity-related pathways, and the most significant pathways under different salinities were the AMPK signaling pathway (ko04152), lysosome (ko04142) and drug metabolism-cytochrome 450 (ko00982). AMPK phosphorylates PFK2 (phosphofructokinase-2), indicating that AMPK is directly involved in the regulation of glycolysis. Therefore, we speculated that under salinity stress, *M. nipponense* might induce ion transport through the activation of ion channels, which requires energy consumption and may be mainly accomplished through glucose metabolism to produce ATP.

In addition, the down-regulated genes were closely related to lipid metabolism. Crustacyanin-like lipocalin is a crustacyanin-like protein that is unique to shrimp and crabs and is involved in the transport of lipids [56]. Apolipophorin is a nutrient storage protein that transports fat. It is synthesized in fat and is mainly used for lipid storage and transport [57]. Monoglyceride lipase is involved in the enzymatic reaction of the decomposition of fatty substances [58]. Delta-9 desaturase is involved in lipid metabolism and catalyzes the formation of mono-chain unsaturated fatty acids [59]. Lipase 3-like isoform is an isomer of lipase which participates in the process of lipid metabolism through the decomposition of fat [60]. Acyl-CoA-binding protein is an essential protein in the metabolism of biological lipids, involved in lipid synthesis and β-oxidation [61]. Hou et al. [62] conducted transcriptomics analysis on *Litopenaeus vannamei* under long-term low-salt stress and found that significant changes had taken place in energy metabolic pathways, especially lipid metabolic pathways, including fatty acid biosynthesis and arachidonic acid metabolism. Lipids have a great influence on energy supply and osmotic pressure regulation when *Litopenaeus vannamei* adapts to a low-salt environment [63]. Studies have reported that when aquatic animals are stressed by a non-isotonic state, they reduce the synthesis of unsaturated fatty acids, change the composition of the cell membrane and reduce the fluidity of cell lipid membrane to maintain the balance of water and salt in their bodies [64]. When *Litopenaeus vannamei* was exposed to low salt, long-chain unsaturated fatty acids were at a significantly higher level than in the control group, and the change in salinity affected the lipid metabolism [65]. The content and composition of fat in the hepatopancreas of *Eriocheir sinensis* also changed due to long-term saltwater culture [66]. Therefore, we speculated that the inhibition of lipid metabolism might be due to the reduction in synthetic fat and changes in some lipid metabolism pathways. Finally, *M. nipponense* was able to up-regulate glucose metabolism, down-regulate lipid metabolism and change the energy supply required for osmotic pressure regulation to maintain homeostasis.

Gill tissue is a direct contact between aquatic animals and the external water environment, and also an important part of ion exchange, which plays an important role in maintaining osmotic pressure balance in animals [25,26]. Studies have shown that shrimp’s gill tissue epithelium, gill filaments, gill lamella and columnar cells are affected under salinity stress, resulting in structural damage to the gill tissue [67]. The observed results of gill tissue slices in this study showed that the gill slices in group S14 were evenly spaced and arranged compared with the control group, with no secretions between the gill slices, uniformly arranged columnar cells, normal connective tissue connections or complete gill tissue structure. The gill structures of the S8 and S22 groups were damaged to different degrees. These results indicate that a salinity of 14 was a relatively suitable salinity for the survival of *M. nipponense*, and was close to the isotonic point. When *M. nipponense* was in an environment of low or high salinity, the gill tissue was damaged. This result is consistent with the previous study of Huang et al. [41], which found that a salinity of 14‰ can promote the growth of *M. nipponense*, and higher salinity conditions may cause physiological damage.

The analysis of enzyme activities in the gill tissues of *M. nipponensis* showed that PK, HK and PFK all increased first and then decreased with increased salinity, and reached the highest activity level at 14‰. PK is involved in glycolysis and catalyzes the formation of a second ATP reaction [29]. HK is a key enzyme in glycolysis and plays a role in regulating sugar metabolism [30]. PFK is an allosteric enzyme that plays a key role in glycolysis pathways and is a rate-limiting enzyme in glycolysis [31]. However, the enzyme activity of LDH was at its lowest at 14. LDH is an important enzyme in anaerobic glycolysis and gluconeogenesis [32]. It can catalyze the conversion between pyruvate and lactic acid. Therefore, when the salinity is 14, a large amount of pyruvate is produced by glycolysis and enters the tricarboxylic acid cycle to produce a large amount of energy, improving the proportion of sugar metabolism in the metabolic energy source and keeping the body of the animal stable in a saline environment. The expression levels of genes related to glucose metabolism showed that with the increase in salinity, the expressions of *HK*, *PK* and *G-6-Pase* showed an overall trend of first increasing and then decreasing. Studies have shown that gills, as key organs for ion transport in crustaceans, can resist salt stress through osmotic regulation, which requires energy consumption partly derived from glucose metabolism. Therefore, shrimp can respond to environmental changes under salinity stress by changing the expression of genes and enzyme activities related to glucose metabolism [47,67]. The expression of *LDH* was the highest in the isotonic salinity group (S14) and was inhibited in the S22 group. Compared with the control group, there was no significant difference in *LDH* expression between the control group and the S14 group, and the *LDH* expression in the S8 and S22 groups was significantly increased compared with the control group. The results were in accordance with the results of enzyme activity detection. Transcriptomic analysis of the shrimp after salinity stress showed that related genes were mainly involved in fatty acid metabolism, glycolysis/gluconeogenesis, glycerophospholipid metabolism, etc. [68,69]. In addition, the activities of immune-related enzymes and metabolic enzymes also showed significant changes, which was consistent with the results of this experiment.

## 5. Conclusions

Transcriptome data of *M. nipponense* were obtained via the RNA-seq technique. Among 593 DEGs, 15 were randomly identified. The results showed that the genes related to ion transport and glucose metabolism were up-regulated and the genes related to lipid metabolism were down-regulated. In addition, the gill sections of the S14 group were evenly spaced and arranged, with normal connective tissue connection and complete gill structure, whereas the gill structures of the S22 groups were damaged. The activities of glucose metabolism-related enzymes PK, HK and PFK first increased and then decreased with an increase in salinity, and reached their highest points at S14. On the contrary, the enzyme activity of LDH was lowest at S14. Changes in the expression of genes encoding these four enzymes were consistent with enzyme activity. This study showed that salinity stress can activate ion transport channels and promote the level of glucose metabolism, which was up-regulated. High salinity levels caused damage to the gill tissue of *M. nipponense*. Overall, the results of this study pave the way for further research on salinization culture of shrimp.

## Figures and Tables

**Figure 1 animals-13-02884-f001:**
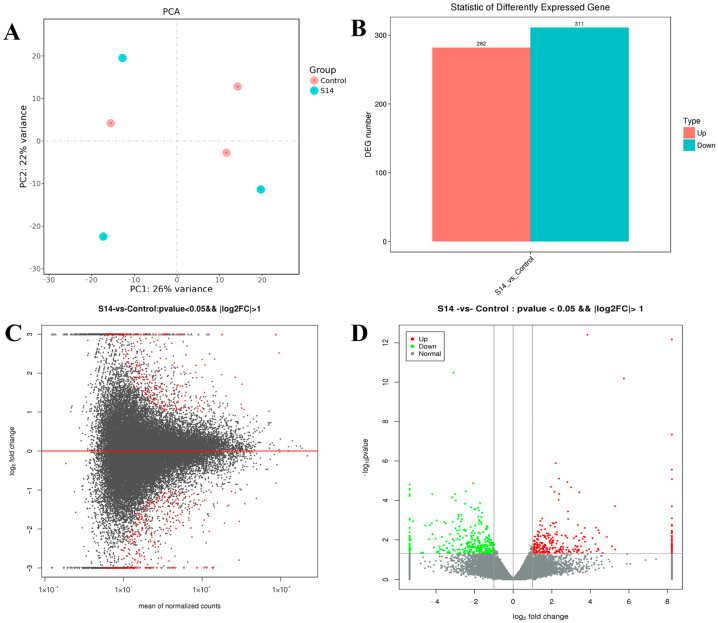
Differentially expressed genes (DEGs) identified in *M. nipponense*. (**A**) Principal component analysis (PCA) plot of the salinity 14‰ (S14) and the control groups. (**B**) Number of DEGs identified in the S14 and the control groups. Up- and down-regulated unigenes are shown in red and cyan, respectively. The *X*-axis shows two comparisons. The *Y*-axis represents the total number of DEGs. (**C**). DEGs between the S14 and the control groups were visualized as an M-versus-A plot (log_2_ fold change [FC] vs. mean of normalized count). Red dots represent transcripts with significant differences shown for unigenes. (**D**). DEGs between the S14 and the control groups are represented on a volcano plot map (log10 *p*-value vs. log_2_FC). Red and green dots indicate transcripts with positive and negative change values, indicating the up-regulation and down-regulation of DEGs.

**Figure 2 animals-13-02884-f002:**
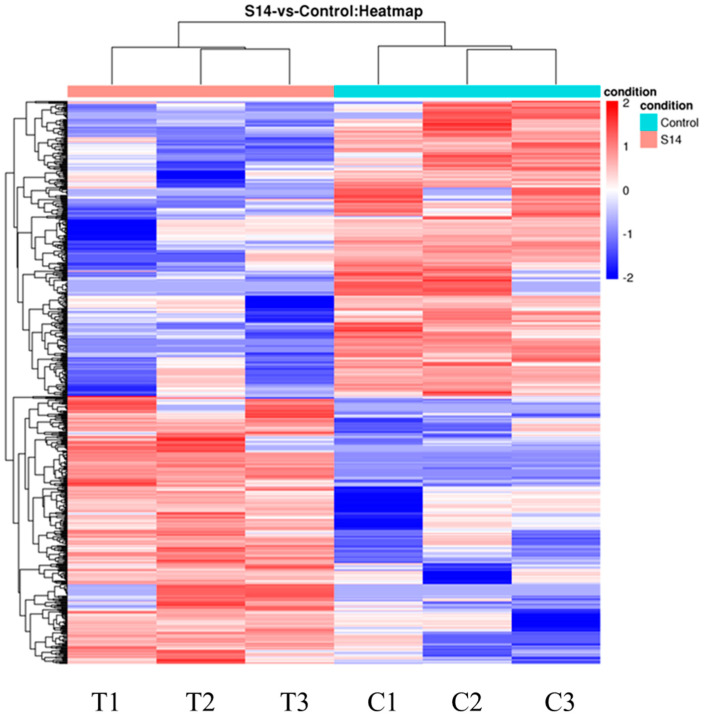
Hierarchical clustering for the DEGs between the S14 (T) and control groups (C). Red represents up-regulation and blue represents down-regulation.

**Figure 3 animals-13-02884-f003:**
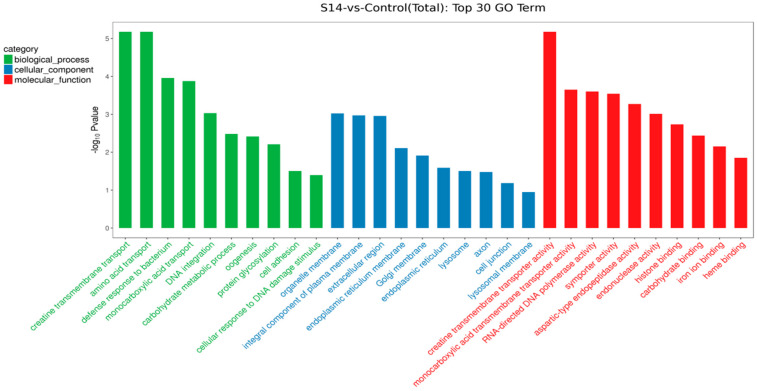
GO enrichment of DEGs. The top 30 GO terms were identified for biological processes, cellular components and molecular functions (*p* < 0.05; the unigene number of GO terms was >2).

**Figure 4 animals-13-02884-f004:**
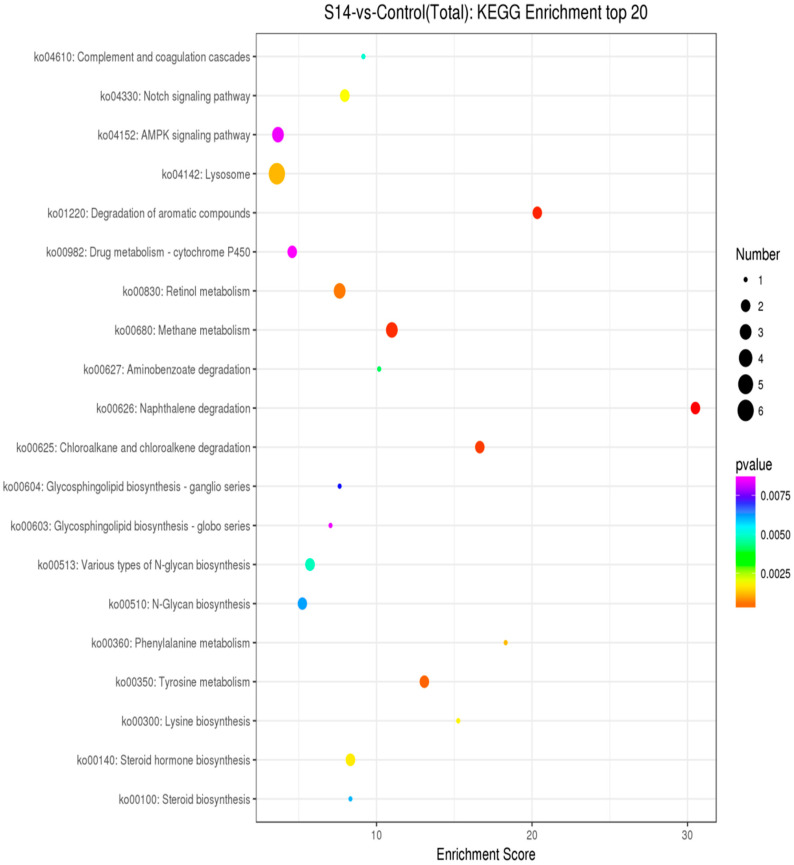
KEGG pathway analysis of DEGs. The *Y*-axis represents pathways, and the *X*-axis represents the enrichment score. The color and size of each bubble represent the enrichment significance and the number of genes enriched in the pathway, respectively (*p* < 0.05).

**Figure 5 animals-13-02884-f005:**
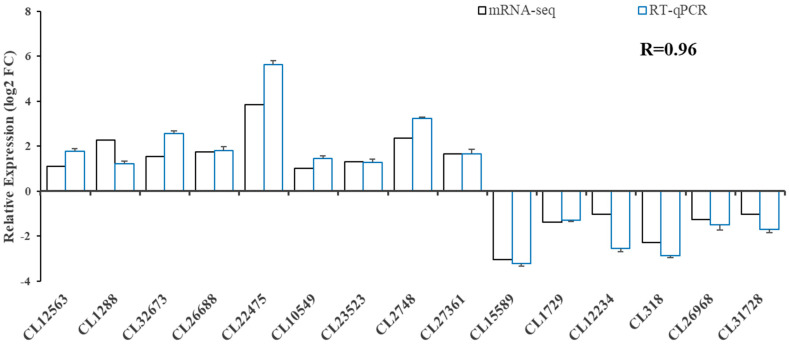
Validation of RNA-seq profiles using real-time qPCR.

**Figure 6 animals-13-02884-f006:**
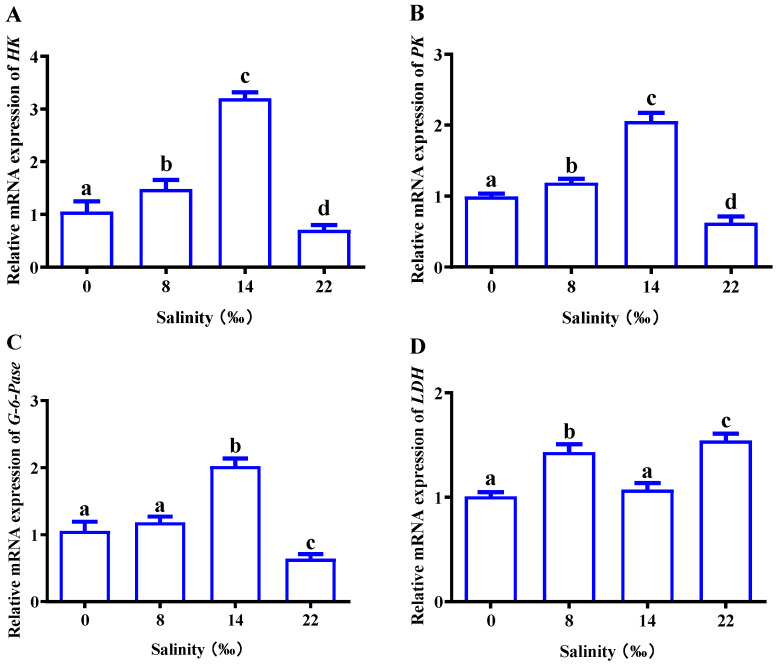
Effects of salinity on the mRNA expressions of (**A**) *HK*, hexokinase; (**B**) *PK*, pyruvate kinase; (**C**) *G-6-Pase*, 6-phosphofructokinase; (**D**) *LDH*, lactate dehydrogenase; in the gills of *M. nipponense*. The data are presented as mean ± SEM values (*n* = 3). Different letters above bars of the same series indicate significant differences (*p* < 0.05) among treatments.

**Figure 7 animals-13-02884-f007:**
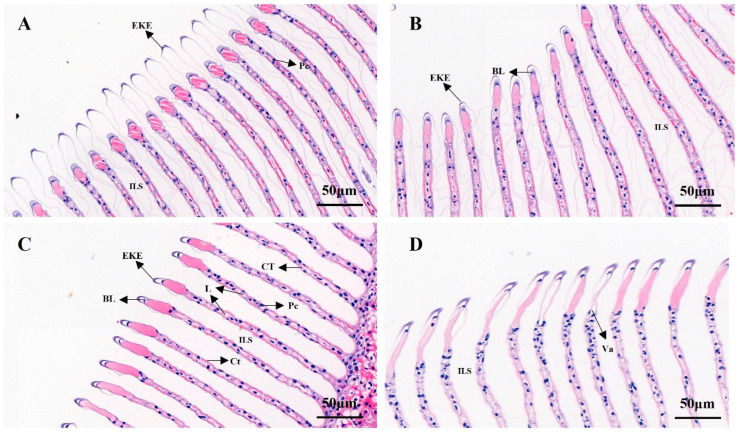
H&E-stained sections of gill tissue at different salinities (‰). (**A**) Control group; (**B**) S8 group; (**C**) S14 group; (**D**) S22 group. L: lamellae; Ct: cuticle; CT: connective tissues; ILS: interlamellar space; Pc: pillar cell nuclei; BL: basal lamina; Va: vacuole; EKE: external keratinous epidermis. Scale bar = 50 µm.

**Figure 8 animals-13-02884-f008:**
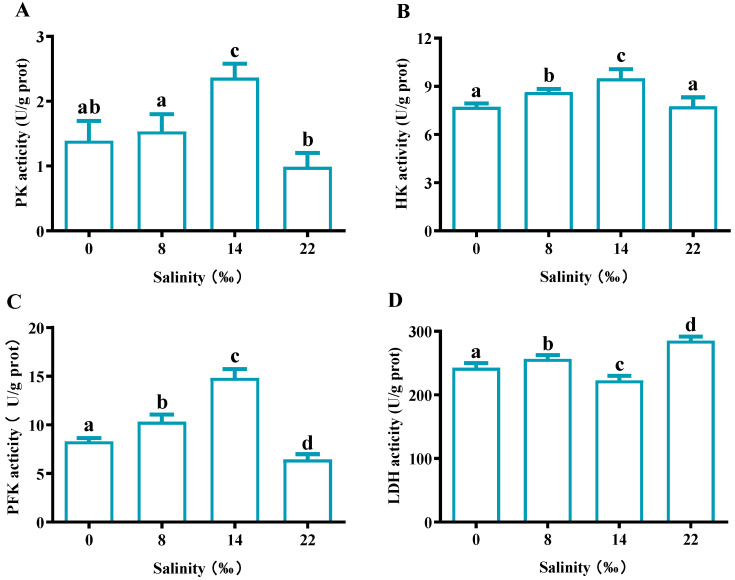
Effects of salinity glycoprotein-metabolism-related enzyme activities in the gills of *M. nipponense*. (**A**) pyruvate kinase (PK) content, (**B**) hexokinase (HK), (**C**) 6-phosphate fructose kinase (PFK) and (**D**) lactate dehydrogenase (LDH). The data are presented as mean ± SEM values (*n* = 3). Different letters above bars of the same series indicate significant differences (*p* < 0.05) among treatments.

**Table 1 animals-13-02884-t001:** Sequences of primers used for qRT-qPCR analysis of transcriptomic validation.

Gene	Description	Sequence (5′-3′)	Up or Down
*CL12563Contig1*	Cytochrome p450	F: AATGGTTGCGAGTCGAAGGT	Up
		R: ACATCTGGGTACTTGGCTGC	
*CL1288Contig1*	Na^+^/K^+^ symporter	F: CGCCTTCGATTTGCATCTGG	Up
		R: TGCACTCGATTCCTGTTGCT	
*CL32673Contig1*	Sodium- and chloride-dependent glycine transporter	F: ATGTACTACAGTGTCGGGCC	Up
		R: GAGGAGCCCATTCTGGAAGA	
*CL26688Contig1*	sodium-dependent proline transporter-like	F: CATGCCGCTCATGTTCTTCG	Up
		R: GGTCACTGATCGCACCAGAA	
*CL22475Contig1*	Sulfatase	F: TCGAACTGGTTACACAGGGC	Up
		R: CTCCTAGTGGTGGTAGGCCT	
*CL10549Contig1*	Monocarboxylate transporter	F: TGGGTCATCGTTTTCGCTGA	Up
		R: AGCCGTGAAGCACAATACCA	
*CL23523Contig1*	Fucosyltransferase	F: GCGACATGAAAACACTGCCA	Up
		R: AAAACCAAATGTGCCGACGG	
*CL2748Contig1*	Mannosyl	F: CGCCTACATCACGCAGAGAT	Up
		R: TTCGCCACCTGACTCTTTCC	
*CL27361Contig1*	C-type lectin	F: ACATGGACCAATGGATCGGG	Up
		R: AGCTGATCCATCGATGTCTCG	
*CL15589Contig1*	Crustacyanin-like lipocalin	F: GTACTAGCAGCCGTCTTGGG	Down
		R: TCGATGCCCATTTCGGTGAA	
*CL1729Contig1*	Apolipophorin	F: GCCAAAGCAAAGTGGGAAGG	Down
		R: GTGGAAGTCGACCTTGCTCA	
*CL12234Contig1*	Monoglyceride lipase	F: GGATATCCCGGGCTTCCTTG	Down
		R: AGCCTCATGACGAACACTGG	
*CL318Contig1*	Delta-9 desaturase	F: GTGTGACAGTCCTGAGGTCG	Down
		R: AACGCACGTCGAGGATTCTT	
*CL26968Contig1*	Lipase 3-like isoform	F: CGAGTCAGCCTAAGAGAGCG	Down
		R: GCCTTGTAAGGGTCCCACAA	
*CL31728Contig1*	Acyl-CoA-binding protein	F: ATGAGGCTGCTGAGAAGGTC	Down
		R: AAGCCTCCATAGCAGCATCC	

**Table 2 animals-13-02884-t002:** Sequences of primers used for qRT-PCR of glucose-metabolism-related genes.

Primer Name	Primer Sequence (5′–3′)	NCBI Database-Gene Accession Number
*G-6-Pase F*	F:CGTGGACCTTTCTTCATTAG	MK307768.1
*G-6-Pase R*	R:ACCATCAACCATTTGAGAAG	
*HK F*	F:TGTTCCCCAGCCGATTATGG	KT932419.1
*HK R*	R:CGGCGCACTTGAATCCTTTG	
*PK F*	F:AGAAACCCAGACCAACCC	KP690140.1
*PK R*	R:TAGTCGCCCTTGGCAGTC	
*LDH F*	F:TCGACATCTTCAAGGGCATC	MF033360.1
*LDH R*	R:AGGTCAAAACGTCAACGGG	
*β* *-actin F*	F:CAGGTCGTGACTTGACCGAT	KY780290
*β* *-actin R*	R:CGTCAGGGAGCTCGTAAGAC	

Note: G-6-Pase: 6-phosphofructokinase; HK: hexokinase; PK: pyruvate kinase; LDH: lactate dehydrogenase; β-actin: reference gene.

**Table 3 animals-13-02884-t003:** Effects of salinity on survival rate, weight gain rate and hepatopancreas index of *M. nipponense*. In the same colunm, values with different letter superscripts represent significant differences (*p* < 0.05).

Parameters	Different Salinity			
	0‰	8‰	14‰	22‰
SR (%)	79.03 ± 3.82 ^ab^	83.60 ± 2.61 ^ab^	87.17 ± 4.30 ^a^	69.91 ± 1.82 ^b^
WGR (%)	26.69 ± 3.95 ^a^	31.14 ± 2.59 ^a^	40.39 ± 5.41 ^b^	15.82 ± 4.46 ^c^
HSI (%)	3.92 ± 0.14 ^a^	3.80 ± 0.60 ^a^	4.76 ± 0.45 ^b^	4.00 ± 0.20 ^a^

**Table 4 animals-13-02884-t004:** Statistics for the sequenced transcriptome data.

Sample	Raw_Reads	Raw_Bases	Clean_Reads	Clean_Bases	Valid_Bases	Q30	GC
S0-1	49,299,632	7,394,944,800	48,187,746	7,047,263,043	95.30%	95.50%	46.06%
S0-2	49,394,440	7,409,166,000	48,186,892	7,052,820,332	95.19%	95.27%	47.26%
S0-3	49,316,632	7,397,494,800	48,230,800	7,052,349,647	95.33%	95.46%	47.44%
S14-1	48,933,058	7,339,958,700	47,935,956	7,009,780,552	95.50%	95.68%	46.58%
S14-2	48,934,960	7,340,244,000	47,982,628	7,007,194,602	95.46%	95.83%	46.89%
S14-3	48,973,892	7,346,083,800	47,969,240	7,018,985,183	95.55%	95.70%	47.55%

## Data Availability

The authors declare that all data supporting the conclusions of this study are available within the article.

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
