# Peer review of "Effects of Salinity Stress on Histological Changes, Glucose Metabolism Index and Transcriptomic Profile in Freshwater Shrimp, Macrobrachium nipponense"

_animals, 2023, doi:10.3390/ani13182884_

Round 1

Reviewer 1 Report

Review for the paper "Effects of salinity stress on histological changes, glycoprotein metabolism index and transcriptome analysis in freshwater shrimp, Macrobrachium nipponensis" by Yiming Li, Yucong Ye, Wen Li, Xingguo Liu, Yunlong Zhao, Qichen Jiang, Xuan Che submitted to "Animals".

General comment.

The oriental river prawn Macrobrachium nipponense, a native crustacean, inhabits various water bodies across China, spanning from the serene rivers to the expansive lakes, reservoirs, and ditches. A significant portion of the Macrobrachium nipponense supply is sourced from its capture fishery. As a flawless denizen of freshwater, this animal splendidly completes its life cycle within the realms of freshwater, wherein the entire process of breeding and nurturing of larvae occurs in vast outdoor ponds. Previous research has shed light on the fact that adult Macrobrachium nipponense can thrive harmoniously within a salinity range of 7–20. In order to unravel the intricate mechanisms that underlie the adverse impacts of heightened salinity on this species, the authors embarked upon a laboratory investigation. Remarkably, the results of their transcriptome sequencing unveiled an upsurge in the expression of genes associated with ion transport and glycoprotein metabolism. Intriguingly, a salinity level of 14 g/L was found to promote the glycoprotein metabolism in Macrobrachium nipponense, while elevated salinity levels detrimentally affected the delicate gill tissue. The authors posited that, when confronted with salinity stress, Macrobrachium nipponense might trigger ion transport through the activation of ion channels, which necessitates energy consumption. This vital process is primarily facilitated through glycoprotein metabolism, ultimately leading to the production of ATP. These findings furnish a solid theoretical framework for comprehending the physiological response mechanism of this extraordinary shrimp species to salinity. In essence, this study serves as a springboard for future investigations into the realm of salinization culture of shrimp. It is worth noting that the authors adhered to standard methodologies in data collection and processing, supplementing the text with pertinent tables and figures. I can recommend this paper for publication after minor revisions.

Recommendations:

1. L 50. The sentence "In recent years…" seems unclear. Please revise to enhance comprehension.

2. L 103. The sentence "M. nipponense, also known…" seems unclear. Please revise to enhance comprehension..

3. L 199. The sentence "Melting curve…" is not adequately clear. Please revise to enhance comprehension.

4. Materials and Methods: To enhance the comprehensibility of the experimental procedures, it is recommended that the authors include additional information regarding the size range, weight, and age of the experimental animals. Furthermore, providing sample sizes for each group would contribute to a more comprehensive understanding of the study. Additionally, details about the rearing conditions, such as temperature, light regime, and biogenic elements, should be incorporated into the text.

5. Results: It is crucial for the authors to include information about the mortality rates observed during the experiment. This will offer valuable insights into the overall outcomes.

6. Fig. 1 and Fig. 2: Upon examining the figures, it is evident that the sample sizes for the comparisons were only three for each group. This seems to contradict the statement made in line 132, which mentions that "nine M. nipponense were randomly selected from each group." Please explain this inconsistency and revise if necessary to ensure accuracy.

7. L 295-298: In the text, it is mentioned that "Fig. 6" should be changed to "Fig. 8." Please review the order of figures and ensure that they are placed after the relevant sections to maintain clarity and coherence.

8. L 309: Verify the order of figures to ensure accuracy and coherence.

9. Discussion. The authors should consider the results by Hongtuo et al. (2012) and Huang et al. (2019)

Hongtuo, F. U., Sufei, J., & Yiwei, X. (2012). Current status and prospects of farming the giant river prawn (M acrobrachium rosenbergii) and the oriental river prawn (M acrobrachium nipponense) in C hina. Aquaculture Research, 43(7), 993-998.

Huang, Y. H., Zhang, M., Li, Y. M., Wu, D. L., Liu, Z. Q., Jiang, Q. C., & Zhao, Y. L. (2019). Effects of salinity acclimation on the growth performance, osmoregulation and energy metabolism of the oriental river prawn, Macrobrachium nipponense (De Haan). Aquaculture Research, 50(2), 685-693.

10. Citations in the text and references should be formatted according to the Rules for Authors.

Specific remarks.

L 4. Consider replacing “Macrobrachium nipponensis” with “Macrobrachium nipponense”

L 18. Consider replacing “The gills was used” with “The gills were used”

L 39. Consider replacing “obvious effect on osmotic pressure” with “an obvious effect on the osmotic pressure”

L 193. Consider replacing “Gill tissues were used to transcriptome validation and analyze” with “Gill tissues were used to for transcriptome validation and to analyze”

L 198. Consider replacing “program for” with “program”

L 222. Consider replacing “Levene test” with “Levene's test”

L 223. Consider replacing “Single factor” with “A one-way”

L 224. Consider replacing “Tukey multiple comparison test” with “the Tukey multiple comparison test”

L 395. Consider replacing “Lipids have great influence” with “Lipids have a great influence”

Minor

Reviewer 2 Report

The manuscript “Manuscript ID: animals-2553405” entitled “Effects of salinity stress on histological changes, glycoprotein metabolism index and transcriptome analysis in freshwater shrimp, Macrobrachium nipponensiss” provides comprehensive scientific results about the effect of different salinities on the gills and physiology of Macrobrachium nipponensiss. The authors used RNA sequencing to discover the molecular pathways and related genes. Moreover, the authors applied histological analysis, and investigated several enzymes in the gills.

In general, the topic is important for the readers of “Animals” and also for aquaculture scientists. The manuscript was well-written and well-structured. I would like to recommend the acceptance of this manuscript after a minor revision and my specific comments are as below:

General comments:

There are some errors in the English language style. There is no significant grammar error, but there is room for some style improvement. The manuscript must undergo revision by a language editing service or a native speaker. There are several unsuitable and informal terms in the manuscript. Examples: Line 71: The term “laws” is not suitable there. Line 106: The term “loved ” is not suitable there. In line 114: What means “acute culture”?

My specific comments are as below:

Title

Please change “transcriptome analysis” to “transcriptomic profile” or some other similar terms.

Abstract:

Line 18: The term “treated with” seems not to be suitable here.

1. Introduction

Line 50-51: This sentence needs improvement. What means “parameter adjustment has also continued to improve”?

Line 54: What means “The adjustment of the aquaculture mode”?

Line 71: The term “laws” is not suitable here. The term “laws” is usually refereeing to the regulation that is made by humans.

Line 77: In line 77 you can give some information about the genes and pathways that helps shrimps and prawns to cope with the salinity stress. Recently several adaptive mechanisms have been found to help shrimps to cope with salinity stress. For example, the renin-angiotensin system (RAS) seems to be important in the salinity adaptation in Litopenaeus vannamei. In the introduction part, you can also mention some mechanisms and pathways that are important in shrimps and prawns to cope with salinity stress.

You can read the below publications as an example or search and find other similar publications:

Comparative Transcriptome Analysis Reveals the Adaptation Mechanism to High Salinity in Litopenaeus vannamei, Front. Mar. Sci., 02 June 2022

The role of the renin-angiotensin system (RAS) in salinity adaptation in Pacific white shrimp (Litopenaeus vannamei). Frontiers in Endocrinology13. 2022.

Line 106: The term “loved ” is not suitable here. In a scientific paper, it is generally best to use more formal and objective language. Some terms such as “favored” can be more suitable.

Line 108: exchange 2 billion yuan into US $.

Line 114: What means “acute culture”?

Line 118: Change “gill tissue” to “gills”

2. Material and methods

Line 122-123: What means “shrimps from Shanghai Xiangsheng Fisheries Cooperative were selected”? Do you mean purchased?

Line 124: domestication? This term usually uses for the process of taming an animal and keeping it as a pet.

Line 126: Please use the same unit for salinity in all parts of the manuscript. Here is ‰ but in the abstract is g/L

Line 128: The term “domestication” is not suitable

Line 138: Only from isotonic groups (14 ± 0.1 ‰)? What about other salinity groups?

Line 139-140: This sentence needs revisions

Line 142-143: What as the criteria for considering the RNA to be high quality and suitable for sequencing?

Line 174: “p-value”. Italic “p

Line 195: I can't see any primer in Table 1

What are the differences between Table 2 and Table 3?

3. Results

Line 229: The location of Table 1 seems to be wrong. Table 1 must be Table 3.

Line 230: I can't find SUB12933053 on the NCBI website.

Line 257: In Figure 2, what is 6_14_3? Write the name of the samples. For example, C1, C2, C3.

Line 273: In Figure 3, Is this GO annotation or GO enrichment? Basically, it is not necessary to perform statistical analysis for GO annotation. But for GO enrichment it is necessary.

Line 281: what was the p-value for KEGG enrichment analysis. Add it to the Figure 4 caption.

Line 297: LDH must be written in italic format. Check all parts of the manuscript.

4. Discussion

Line 353-355: This sentence doesn't have a strong support. This study (Daniel et al., 2022) is related to pharmacology studies in humans. While there is many research showing the role of Cytochrome p450 in the salinity adaptation in shrimps.

 In shrimps, Cytochrome p450 plays an important role in the energy metabolism and oxidative stress under salinity stress. You can take on the below paper as example or search to find more publications. 

Evidence from transcriptome analysis unravelled the roles of eyestalk in salinity adaptation in pacific white shrimp (Litopenaeus vannamei). General and Comparative Endocrinology329, p.114120.

Line 422: “14” What is the unit?

There are some errors in the English language style. There is no significant grammar error, but there is room for some style improvement. The manuscript must undergo revision by a language editing service or a native speaker. There are several unsuitable and informal terms in the manuscript. Examples: Line 71: The term “laws” is not suitable there. Line 106: The term “loved ” is not suitable there. In line 114: What means “acute culture”?

Reviewer 3 Report

The manuscript entitled "Effects of salinity stress on histological changes, glycoprotein metabolism index and transcriptome analysis in freshwater shrimp, Macrobrachium nipponensis" wrote by Yiming Li, et al. is interesting and results can help us further understand the effects of salinity stress on shrimp, and have a positive effect on shrimp culture. The authors need address some important issues as the following, which may be helpful to improve the ms.

1.    In the part of “Abstract,” there was no sample information in transcriptome analysis.

2.    Line 30, the abbreviation of enzymes and genes should be written in the full name of enzymes and genes.

3.    Line 106, please delete “have been”.

4.    Line 113, please change “growth” into “metabolic physiology”.

5.    Line 114, 397, “have reported” should be “have been reported”.

6.    Line 126, please cite references stating that salinity 14 is isotonic.

7.    Line 138, why did only one group (14 ± 0.1 ‰) in the experimental group for transcriptome sequence? I guess that it was determined by the result of histological analysis. So, in the part of “Material and methods”, it is better to write the histological analysis firstly, and then introduce the transcriptome sequence. In this case, it will be also corresponding to the Title.

8.    Line 173, please check “p” into italic.

9.    Please check whether it is “glucose metabolism” or “glycoprotein metabolism” that keeps the full text consistent.

10. Line 197-198, the unit of volume was miswritten.

11. Line 296-298. The Fig. 6…should be Fig. 8…, please notice the corresponding icon in the text and Figures. The same mistake was also made in the following Line 310-324. In addition, the order of Figure number in the result part was also incorrect.

12. Line 327, please delete “content”.

13. Line 343-347, please simplify this description

14. Line 347-351, please revise this part of the description.

15. Line 396, please change“ADAPT”into “adapt”.

16. Line 409-419, please add more discussion to the description and cite literature explaining how salinity affects morphological changes in gill tissue.

17. Line 434-439, more discussion need to add the description and cite literature explaining how salinity affect change in glucose metabolism.

18. In the part of “Results”, the results in the Line 291-298, there was no subtitle about this paragraph.

19. The format of references need to be uniformed.

Overall, the language is clear, but some areas need further improvement.

Reviewer 4 Report

The proposed manuscript presents an interesting analysis of the changes caused at molecular and histological levels due to salinity exposure in freshwater shrimp with commercial value. An actual topic due to the salinization of inland waters due to several anthropogenic and natural factors.

However, some revisions should be done before publication.

The authors argued that studies assessing transcriptomics under salinity changes are relatively scarce. I disagree with this statement, as recent years have witnessed a growing body of literature delving into this topic. Several papers from 2023 have specifically investigated the impact of salinity on shrimps, such as those cited below. The comparison and interpretation of data appear skewed, with a disproportionate emphasis on fish-related studies over those involving shrimps, ensuring a comprehensive analysis that adequately reflects the available literature. Consequently, I propose that the authors revise their discussion to encompass these new findings, thereby enriching the depth of their analysis.

Example

Fan et al 2023 Transcriptome, Proteome, Histology, and Biochemistry Analysis of Oriental River Prawn Macrobrachium nipponense under Long-term Salinity Exposure

Zhang et al 2023 Histological, Physiological and Transcriptomic Analysis Reveal the Acute Alkalinity Stress of the Gill and Hepatopancreas of Litopenaeus vannamei 

LL 95: LDH stands for lactate dehydrogenase. Please correct it.

Minor editing of English language required
